# A Meta-Analysis of the Utility of Preoperative Intravenous Paracetamol for Post-Caesarean Analgesia

**DOI:** 10.3390/medicina55080424

**Published:** 2019-07-31

**Authors:** Qin Xiang Ng, Wayren Loke, Wee Song Yeo, Kelvin Yong Yan Chng, Chin How Tan

**Affiliations:** 1MOH Holdings Pte Ltd., 1 Maritime Square, Singapore 099253, Singapore; 2Department of Women’s Anaesthesia, KK Women’s and Children’s Hospital, 100 Bukit Timah Rd, Singapore 229899, Singapore; 3Department of Anaesthesiology, Singapore General Hospital, Outram Rd, Singapore 169608, Singapore; 4National University Hospital, National University Health System, Singapore 119074, Singapore

**Keywords:** pre-emptive, preoperative, paracetamol, pregnancy, caesarean, analgesia

## Abstract

*Background and objectives:* Worldwide, the number of caesarean sections performed has increased exponentially. Some studies have reported better pain control and lower postoperative requirements for opioids when intravenous (IV) paracetamol was administered preoperatively. This meta-analysis thus aimed to investigate the utility of preoperative IV paracetamol for post-caesarean analgesia. *Materials and Methods*: By using the keywords (paracetamol OR acetaminophen) AND [cesarea* OR caesarea* OR cesaria* OR caesaria*], a systematic literature search was conducted using PubMed, Medline, Embase, Google Scholar and ClinicalTrials.gov databases for papers published in English between January 1, 1960 and March 1, 2019. Grey literature was searched as well. *Results*: Seven clinical trials were reviewed, while five randomized, placebo-controlled, double-blind studies were included in the final meta-analysis. Applying per-protocol analysis and a random-effects model, there was a significant reduction in postoperative opioid consumption and pain score in the group that received preoperative IV paracetamol, compared to placebo, as the standardized mean difference (SMD) were −0.460 (95% CI −0.828 to −0.092, *p* = 0.014) and −0.719 (95% CI: −1.31 to −0.13, *p* = 0.018), respectively. However, there was significant heterogeneity amongst the different studies included in the meta-analysis (I^2^ = 70.66%), perhaps owing to their diverse protocols. Some studies administered IV paracetamol 15 min before induction while others gave it before surgical incision. *Conclusion*: This is the first review on the topic. Overall, preoperative IV paracetamol has convincingly demonstrated useful opioid-sparing effects and it also appears safe for use at the time of delivery. It should be considered as a component of an effective multimodal analgesic regimen. Future studies could be conducted on other patient groups, e.g., those with multiple comorbidities or chronic pain disorders, and further delineate the optimal timing to administer the drug preoperatively.

## 1. Introduction

Worldwide, caesarean delivery is an increasingly popular choice amongst expectant mothers, and the number of caesarean sections performed has increased exponentially from approximately 16 million (12.1% of all births) in 2000 to 29.7 million (21.1% of all births) in 2015 [1]. With its increasing prevalence, there is a need to evaluate and optimize the anaesthetic techniques used, especially that of analgesia. 

Opioids have been the mainstay of perioperative analgesia management, however, they are generally avoided in the preoperative and intraoperative period for caesarean sections as opioids are able to cross the placenta and can have adverse effects on the fetus [2]. Opioids can also cause respiratory depression, nausea and vomiting. Moreover, countries like the United States have a growing opioid use disorder epidemic, with many patients’ first encounter with opioids being in the perioperative period [3,4]. Suboptimal management of post-caesarian section pain can also lead to persistent pain in the postpartum period, hindering the care and feeding of the newborn and increasing the risk for opioid dependence and abuse, especially in susceptible individuals. 

Therefore, there is a role for evaluating better pain control strategies and non-opioid analgesic alternatives. A possibility is the use of pre-emptive intravenous (IV) paracetamol. Studies conducted on nonobstetric populations have reported better pain scores and lower postoperative requirements for opioids when one gram of IV paracetamol was administered preoperatively [5,6]. The exact analgesic mechanism of action of paracetamol remains unclear but is thought to have a central effect via descending serotonergic pathways and inhibition of a third COX isoenzyme (designated COX-3) and prostaglandin synthesis [7]. It is hypothesized that central blockade of pain receptors prior to surgery may disrupt the normal conduction of noxious stimuli and reduce the overall perception of pain postoperatively, thereby reducing postoperative analgesia requirements. 

As no previous systematic review or meta-analysis has been done to investigate the utility of preoperative IV paracetamol for post-caesarean analgesia, this study aims to provide the first review on the topic and generate hypotheses for future research. The primary study objective was to examine if patients who received one gram of IV paracetamol preoperatively required fewer opioids in the postoperative period. This would be beneficial for both the mother and her breastfed newborn.

## 2. Materials and Methods

A systematic literature search was conducted in accordance with the Preferred Reporting Items for Systematic Reviews and Meta-Analyses (PRISMA) guidelines [8] for reporting. By using the following combinations of broad Major Exploded Subject Headings (MesH) terms or text words (paracetamol OR acetaminophen) AND (cesarea* OR caesarea* OR cesaria* OR caesaria*), a preliminary search on the PubMed, Medline, Embase, Google Scholar and ClinicalTrials.gov yielded 14,955 papers published in English between 1 January 1960 and 1 March 2019. Grey literature was searched for using the Google search engine and the Open System for Information on Grey Literature in Europe (OpenSIGLE) database. The study investigators (QX Ng, WR Loke and WS Yeo) performed independent title/abstract screening in order to identify articles of interest. For relevant abstracts, full articles were obtained, reviewed and also checked for references of interest. If necessary, the authors of the articles were contacted to provide additional data. 

Full articles were reviewed by three investigators (QX Ng, WR Loke and WS Yeo) for inclusion. Any disagreement was resolved by discussion and consensus amongst the three investigators. The inclusion criteria for this review were as follows: (1) Published randomized, controlled trial, (2) a specified dose of paracetamol was administered preoperatively as an active intervention, and (3) postoperative opioid consumption was quantified. The methodological quality of the eligible clinical trials was assessed using the bias domains described in the Cochrane Handbook for Systemic Reviews of Interventions, version 5.1.0 [9], by discussion and consensus amongst three study investigators. Information on the study design, study population and main conclusions of the studies reviewed are summarized in Table 1.

The primary outcome measure of interest was Cohen’s d, the calculated standardized mean difference (SMD) in postoperative opioid consumption between the intervention group and the placebo group. We also compared the mean postoperative pain score (immediate or in recovery), based on a visual analog scale, between the two groups. Estimates were pooled and the corresponding 95% confidence intervals (95% CI) and P-values were calculated. Heterogeneity amongst the different studies pooled was assessed using the I^2^ statistic and Cochran’s Q test. If heterogeneity was small (I^2^ ≤ 50%), a fixed-effects model was applied for the meta-analysis. All statistical analyses were performed using MedCalc Statistical Software version 14.8.1 (MedCalc Software bvba, Ostend, Belgium; http://www.medcalc.org; 2014) and maintained at a significance level of 0.05 (*p* ≤ 0.05).

## 3. Results

The literature search and abstraction process is summarized in Figure 1. Table 1 shows the salient details of the seven clinical studies reviewed. Two studies were excluded from the final meta-analysis as one was unblinded while the other administered both paracetamol and diclofenac as an active intervention.

Most of the available trials were randomized, placebo-controlled, double-blind trials with a generally low risk of bias (Table 2). However, one study was unblinded and the method for allocation concealment and blinding were unclear in some studies.

As seen in Figure 2, applying a per-protocol analysis and a random-effects model, there was a significant difference in postoperative opioid consumption between the group that received preoperative IV paracetamol and the placebo group as the pooled SMD (Cohen’s d) was −0.460 (95% CI: −0.828 to −0.092, *p* = 0.014). However, there was a significant degree of heterogeneity (Table 3) amongst the different studies included in the meta-analysis (I^2^ = 70.66%), perhaps owing to their diverse protocols. Some studies administered IV paracetamol 15 min before induction while others gave it before surgical incision.

In terms of postoperative pain experienced by the patients in the intervention and control group, the pain scores (based on the visual analog scale for pain) were significantly lower in the group that received preoperative IV paracetamol, compared to the placebo group (SMD −0.719, 95% CI: −1.31 to −0.13, *p* = 0.018). The forest plot analysis of postoperative pain scores was shown in Figure 3.

With regard to the possibility of publication bias, as seen in Figure 4, the funnel plot shows a roughly symmetrical distribution of studies and Egger test was not significant for publication bias (*p* = 0.195). However, the test may be unreliable given the small number of studies available (<10).

## 4. Discussion

Overall, current studies suggest that preoperative IV paracetamol significantly reduced postoperative pain and opioid consumption in the postoperative period. Theoretically, pre-emptive IV paracetamol provides more effective postoperative pain control by preventing peripheral sensitization and disrupting the normal conduction of perioperative noxious stimuli to the medulla spinalis [17]. 

There may be other useful benefits to the administration of preoperative IV paracetamol as well. Laryngoscopy and endotracheal intubation can increase arterial blood pressure and heart rate, posing additional risks, particularly in patients with uncontrolled hypertension [18]. As seen in two studies [11,15], there were significantly blunted heart rate changes following endotracheal intubation in patients who received pre-emptive IV paracetamol before induction. However, this remains speculative and may be partly explained by anachronisms in the particular methodologies of the reported studies. One study also reported improved intraoperative hemodynamic stability, especially before delivery of the neonate [12]. A 2013 meta-analysis of the use of preoperative IV paracetamol in both obstetric and nonobstetric populations also found reduced postoperative nausea and vomiting [19].

Importantly, the use of preoperative IV paracetamol was not associated with an increased incidence of patient side effects [12,13] or adverse effects on the newborn. Administration of preoperative IV paracetamol did not result in elevated neonate cord blood levels [16] and there were no significant differences in mean 1 min and 5 min Apgar scores [11,13].

In terms of the possible mechanisms accounting for preoperative paracetamol’s analgesic effects, it is likely due to its central effects mediated via cyclooxygenase inhibition, serotonergic activation and endocannabinoid enhancement [7,20]. The pharmacokinetics of paracetamol also support its use as a preventive analgesia as the median time to the maximum concentration of paracetamol is 15 min and its cerebrospinal fluid (CSF) concentration is known to lag behind its plasma concentration [21], albeit it remains unclear at what CSF concentration the analgesic effect of paracetamol occurs.

Last but not least, the limitations of the present study should be discussed. First, the present analysis could not account for the anesthetic technique used for surgery (general versus neuraxial; intra-and post-op opiates, either systemic or neuraxial; use and timing of other analgesics), which invariably affects the proper assessment of the contribution of paracetamol to the overall analgesic effect. Three of the studies involved general anesthesia and two involved spinal anaesthesia. In both spinal anaesthetic studies, the parturient women received intrathecal morphine, which would affect postoperative analgesic requirements and render it difficult to interpret findings. The results could be statistically but not clinically significant [22]. Second, most of the available clinical trials recruited patients who were young (between 18 and 40 years of age), generally healthy, with ASA physical status I and II and undergoing elective cesarean surgery. It is thus unclear if the findings can be readily extrapolated to other patient groups. Limited inferences can be made about expectant mothers with multiple comorbidities or those with chronic pain disorders. Third, as the studies had diverse protocols with different timing of IV administration, the optimal timing to administer IV paracetamol should be further investigated and fine-tuned in future studies. Fourth, there are emerging concerns regarding prenatal paracetamol exposure and its potential effect on child neurodevelopment [23]. Although administration of preoperative IV paracetamol did not result in elevated neonate cord blood levels [16] and there were no significant differences in mean 1 min and 5 min Apgar scores [11,13], its long term effects on child neurodevelopment remain unknown. 

Nonetheless, this meta-analysis may bolster the level of evidence supporting the perioperative use of paracetamol in caesarean sections from Level II to Level I. Despite the limitations discussed, the opioid-sparing effect from preoperative IV paracetamol as compared to placebo is convincing enough, albeit differences in the timing of administration.

## 5. Conclusions

This is the first systematic review and meta-analysis to investigate the utility of preoperative IV paracetamol for post-caesarean analgesia. This meta-analysis may bolster the level of evidence supporting the perioperative use of paracetamol in caesarean sections from Level II (unblinded, randomized trials) to Level I (double-blind, placebo-controlled, randomized trials). Random-effects meta-analysis of five randomized, placebo-controlled, double-blind trials found a significant reduction in postoperative opioid consumption in the group that received preoperative IV paracetamol, compared to placebo (pooled SMD −0.460, 95% CI: −0.828 to −0.092, P = 0.014). Overall, paracetamol has useful opioid-sparing effects and it also appears safe for use at the time of delivery. Preoperative IV paracetamol should therefore be considered as a component of an effective multimodal analgesic regimen.

## Figures and Tables

**Figure 1 medicina-55-00424-f001:**
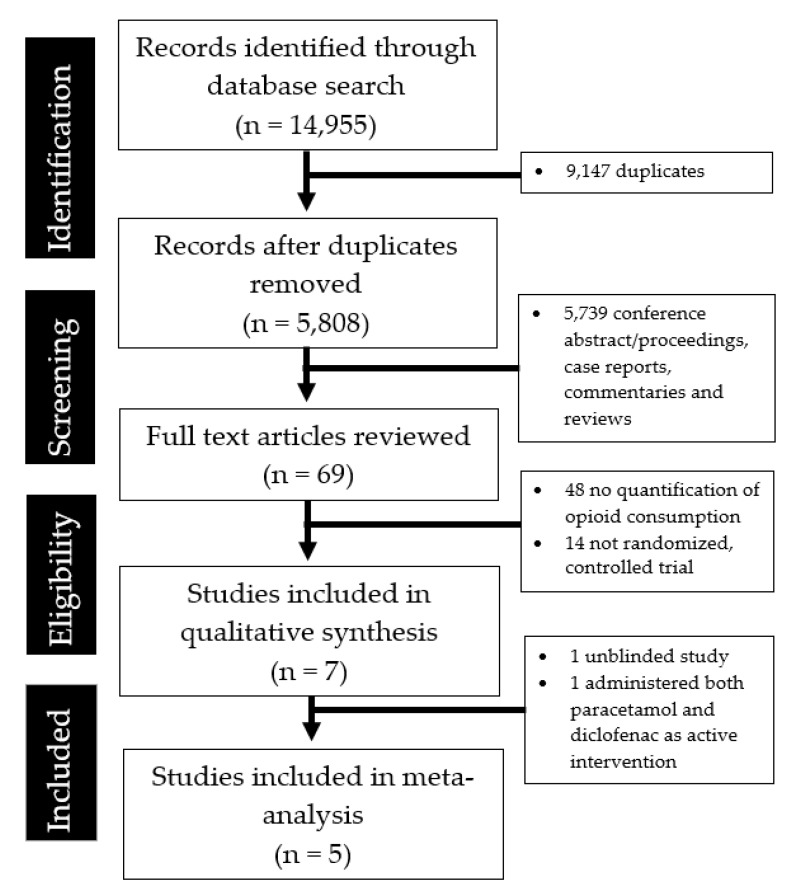
PRISMA flowchart summarizing the literature search and abstraction process.

**Figure 2 medicina-55-00424-f002:**
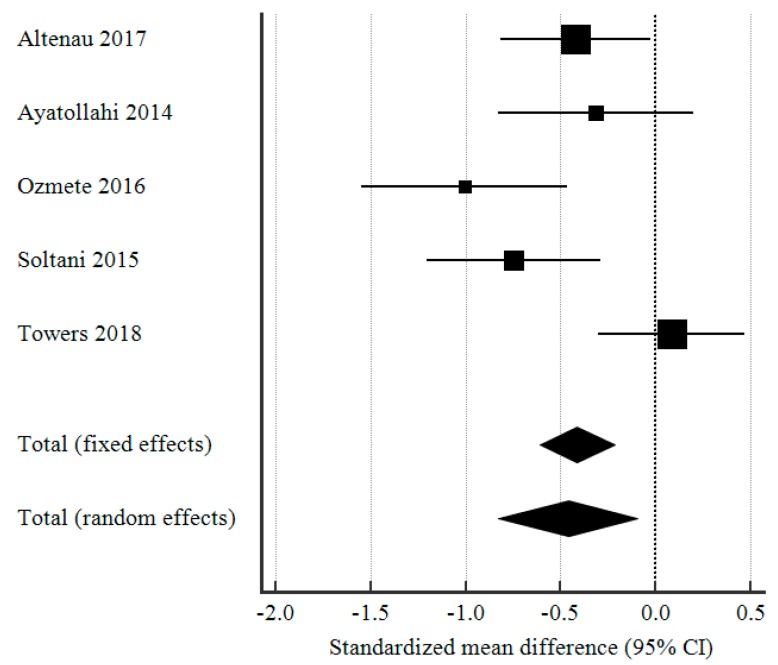
Forest plot showing the standardized mean difference (SMD) in postoperative opioid consumption, comparing patients who received preoperative IV paracetamol and placebo.

**Figure 3 medicina-55-00424-f003:**
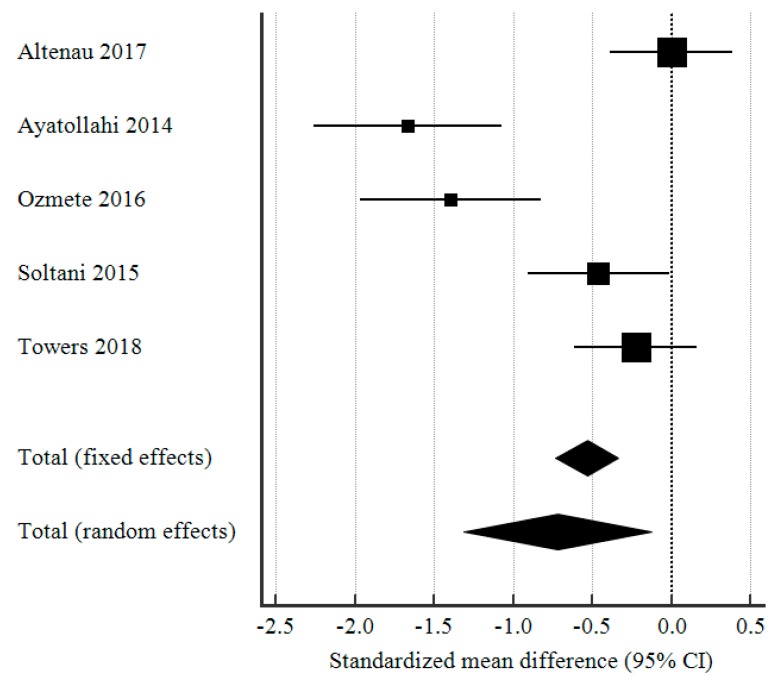
Forest plot showing the standardized mean difference (SMD) in postoperative pain scores (based on visual analog scale) in the immediate postop or recovery period, comparing patients who received preoperative IV paracetamol and placebo.

**Figure 4 medicina-55-00424-f004:**
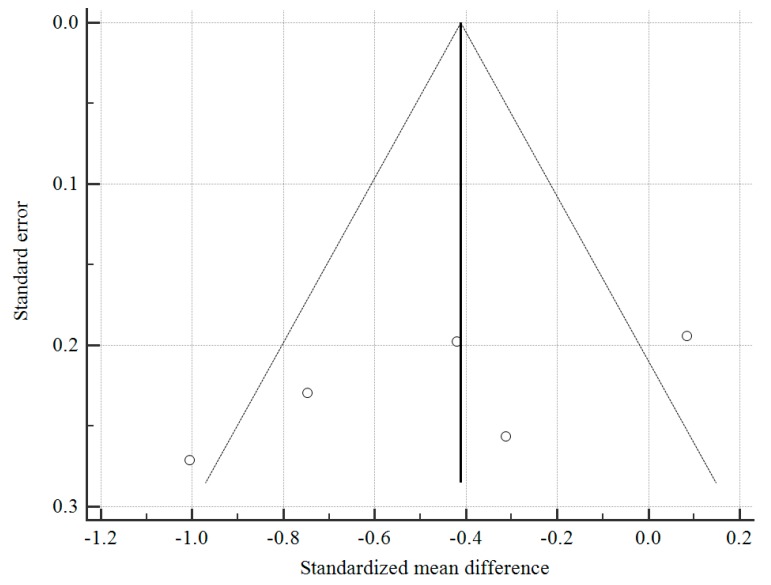
Funnel plot to assess publication bias. Egger test for publication bias = −8.67, 95% CI: −25.2 to 7.94, *p* = 0.195.

**Table 1 medicina-55-00424-t001:** Characteristics of all studies included in this review (arranged alphabetically by first Author’s last name).

Author, Year	Country of Origin	Study Design	Study Sample	Type of Anaesthesia	Intervention	Conclusions
Altenau, 2017 [10]	United States	Randomized, placebo-controlled, double-blind trial	n = 104, pregnant women, scheduled for elective caesarean section, Mean Age 29.6 years	Spinal	- IV paracetamol 1 g given within 30 to 60 min of the surgical incision, and every 8 h for 48 h, for a total of 6 doses	- No significant difference in pain scores but significantly reduced postoperative requirement for opioid.
Ayatollahi, 2014 [11]	Iran	Randomized, placebo-controlled, double-blind trial	n = 60, pregnant women, ASA class I, scheduled for elective caesarean section, Age 18 to 40 years	General	- IV paracetamol 1 g given 20 min before induction	- Improved haemodynamic stability after laryngoscopy and intubation. - Lower requirement for postoperative opioid and later first analgesic request. - No significant difference in mean 1-min and 5-min Apgar scores of newborns.
Hassan, 2014 [12]	Saudi Arabia	Randomized, two-arm, prospective, unblinded trial	n = 58, pregnant women, ASA class I and II, scheduled for elective caesarean section, Age 18 to 39 years	General	- IV paracetamol 1 g given over 15–20 min, 30 min before induction - IV paracetamol 1 g given over 15–20 min, 30 min before the end of the operation	- Patients who received preoperative paracetamol had better hemodynamic stability, especially before delivery of the baby. - They also had lower requirements for intra- and postoperative opioids, longer duration of next analgesia needed and lower incidence of postoperative side effects.
Ozmete, 2016 [13]	Turkey	Randomized, placebo-controlled, double-blind trial	n = 60, pregnant women, ASA class I and II, scheduled for elective caesarean section, Age 18 to 40 years	General	- IV paracetamol 1 g given 15 min before induction	- Significantly reduced postoperative pain and opioid consumption within 24 h after caesarean section. - No significant difference in Apgar scores and patient side effects.
Prasanna, 2010 [14]	Oman	Randomized, two-arm, prospective, blinded trial	n = 80, pregnant women, ASA class I and II, scheduled for elective caesarean section, Mean Age 30.51 years	General	- IM diclofenac sodium 75 mg and IV paracetamol 1 g after induction, before surgical incision - IM diclofenac sodium 75 mg and IV paracetamol 1 g at the end of surgery	- Patients who received pre-incision analgesia had significantly fewer occurrences of incidental pain and reduced postoperative opioid requirements.
Soltani, 2015 [15]	Iran	Randomized, placebo-controlled, double-blind trial	n = 80, pregnant women, ASA class I and II, admitted for urgent caesarean section, Mean Age 28.49 ± 4.63 years	General	- IV paracetamol 15 mg/kg given 15 min before induction	- Significantly blunted heart rate changes following endotracheal intubation and reduced early postoperative pain. - Significantly lower requirements for intra- and postoperative opioids.
Towers, 2018 [16]	United States	Randomized, placebo-controlled, double-blind trial	n = 105, pregnant women, scheduled for elective caesarean section, Mean Age 27.1 ± 2.9 years	Spinal	- IV paracetamol 1 g given 15 min before surgical incision	- No difference in postoperative opioid requirements and length of stay postdelivery. - Administration of IV paracetamol did not result in elevated neonate cord blood paracetamol levels

Abbreviations: American Society of Anaesthesiologists, ASA; intramuscular, IM; intravenous, IV.

**Table 2 medicina-55-00424-t002:** Results of Cochrane Collaboration’s tool for assessing the risk of bias.

Study (Author, Year)	Sequence Generation	Allocation Concealment	Blinding	Incomplete Outcome Data	Selective Outcome Reporting	Other Bias
Altenau, 2017 [10]	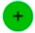	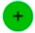	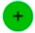	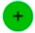	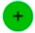	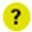
Ayatollahi, 2014 [11]	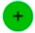	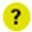	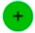	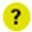	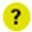	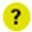
Hassan, 2014 [12]	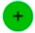	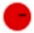	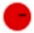	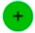	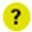	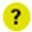
Ozmete, 2016 [13]	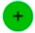	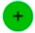	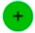	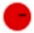	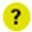	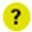
Prasanna, 2010 [14]	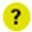	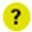	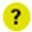	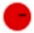	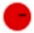	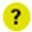
Soltani, 2015 [15]	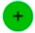	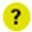	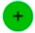	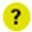	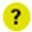	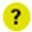
Towers, 2018 [16]	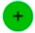	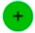	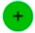	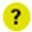	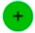	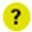

Key: 
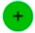
 low risk of bias; 
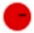
 high risk of bias; 
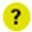
 unclear risk of bias.

**Table 3 medicina-55-00424-t003:** Test for heterogeneity.

Q	13.64
DF	4
Significance level	P = 0.0086
I^2^ (inconsistency)	70.66%
95% CI for I^2^	25.43 to 88.46

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
