# Peer review of "A Meta-Analysis of the Utility of Preoperative Intravenous Paracetamol for Post-Caesarean Analgesia"

_medicina, 2019, doi:10.3390/medicina55080424_

Round 1

Reviewer 1 Report

This is an appropriately conducted meta-analysis evaluating the effects of IV acetaminophen given to patients having a cesarean delivery.  The primary outcome evaluated was postoperative opioid consumption.  The methodology was appropriate for the study. The authors found a consistent trend of less postoperative opioid consumption in the groups that received intravenous acetaminophen. They stated in the table that some studies showed a difference in pain scores afterward. It would be useful if the authors included a Forest plot analysis of pain scores also.

In my opinion, the authors should limit their discussion to opioid use and pain scores. There is not enough data in the papers evaluated to discuss issues like timing of the acetaminophen or the effect on blood pressure and heart rate. The data set is so limited and the study populations so heterogenous no further conclusions can be discussed. For example, the authors state: “Timing appears critical as studies that administered IV paracetamol before induction tended to have significantly reduced pain and lowered opioid requirements in the postoperative period, compared to studies that only administered paracetamol before surgical incision.” This is an interesting point, but the two studies with the highest opioid consumption were also the studies where the surgeries were performed under spinal anesthesia and the patients received IT morphine. Several studies have shown that patients that receive IT morphine have less pain and consume less opioids than patients that do not.  It is hard to imagine that if the patient populations across these studies are similar, that administering IV acetaminophen before induction has a more profound of effect on post-operative pain than IT morphine.  This suggest a large amount of variability in the study populations and these types of additional statements should not be made.

In summary, the authors should limit their the presentation of their findings to postoperative opioid use and pain relief.

Author Response

1. This is an appropriately conducted meta-analysis evaluating the effects of IV acetaminophen given to patients having a cesarean delivery.  The primary outcome evaluated was postoperative opioid consumption.  The methodology was appropriate for the study. The authors found a consistent trend of less postoperative opioid consumption in the groups that received intravenous acetaminophen. They stated in the table that some studies showed a difference in pain scores afterward. It would be useful if the authors included a Forest plot analysis of pain scores also.

REPLY: Thank you for the comment. We have added a forest plot analysis of postoperative pain score (see Figure 3) and added elaboration on this in the manuscript.

“In terms of postoperative pain experienced by the patients in the intervention and control group, the pain scores (based on the visual analog scale for pain) were significantly lower in the group that received preoperative IV paracetamol, compared to the placebo group (SMD -0.719, 95% CI: -1.31 to -0.13, P=0.018). The forest plot analysis of postoperative pain scores was shown in Figure 3.”

2. In my opinion, the authors should limit their discussion to opioid use and pain scores. There is not enough data in the papers evaluated to discuss issues like timing of the acetaminophen or the effect on blood pressure and heart rate. The data set is so limited and the study populations so heterogenous no further conclusions can be discussed. For example, the authors state: “Timing appears critical as studies that administered IV paracetamol before induction tended to have significantly reduced pain and lowered opioid requirements in the postoperative period, compared to studies that only administered paracetamol before surgical incision.” This is an interesting point, but the two studies with the highest opioid consumption were also the studies where the surgeries were performed under spinal anesthesia and the patients received IT morphine. Several studies have shown that patients that receive IT morphine have less pain and consume less opioids than patients that do not.  It is hard to imagine that if the patient populations across these studies are similar, that administering IV acetaminophen before induction has a more profound of effect on post-operative pain than IT morphine.  This suggest a large amount of variability in the study populations and these types of additional statements should not be made.

In summary, the authors should limit their presentation of their findings to postoperative opioid use and pain relief.

REPLY: Thank you for the comments. We agree with the reviewer and have now omitted the statement, “Timing appears critical as studies that administered IV paracetamol before induction tended to have significantly reduced pain and lowered opioid requirements in the postoperative period, compared to studies that only administered paracetamol before surgical incision.” Our study findings are now limited to postoperative opioid use and pain scores.

Reviewer 2 Report

The review is well written, the subject is interesting and very current. I ask the authors to check only grammatical and typing errors.

Author Response

The review is well written, the subject is interesting and very current. I ask the authors to check only grammatical and typing errors.

REPLY: Thank you for the positive feedback. We, the investigators, have enjoyed the process of researching this area of study. We have also checked for grammatical and typing errors as per your suggestion. Hope all is in order.